# Quality-of-Life Assessment in Patients Undergoing Mastectomy and Breast Reconstruction for Moderate-Penetrance Gene-Related Breast Cancer

**DOI:** 10.3390/jcm14041140

**Published:** 2025-02-10

**Authors:** Andreea Cătană, Irina Iordănescu, Gheorghe Gerald Filip, Simona Filip, Mariela Sanda Militaru, Andrada-Adelaida Pătrășcanu, Lorin-Manuel Pîrlog

**Affiliations:** 1Department of Molecular Sciences, Faculty of Medicine, University of Medicine and Pharmacy “Iuliu Hațieganu”, 400012 Cluj-Napoca, Romania; catanaandreea@gmail.com (A.C.); mariela.militaru@reginamaria.ro (M.S.M.); 2Department of Oncogenetics, Institute of Oncology, “Prof. Dr. I. Chiricuță”, 400015 Cluj-Napoca, Romania; 3Regional Laboratory Cluj-Napoca, Department of Medical Genetics, Regina Maria Health Network, 400363 Cluj-Napoca, Romania; 4Genetic Centre Laboratory, Department of Medical Genetics, Regina Maria Health Network, 011376 Bucharest, Romania; irina.iordanescu@reginamaria.ro; 5Ponderas Academic Hospital Bucharest, 014142 Bucharest, Romania; gerald.filip@reginamaria.ro (G.G.F.); simona.filip@reginamaria.ro (S.F.)

**Keywords:** quality-of-life assessment, prophylactic mastectomy, breast reconstruction surgery, breast cancer, moderate-penetrance genes, low-penetrance genes, genetic factors

## Abstract

**Background**. Breast cancer remains a leading cause of cancer-related death among women, with genetic mutations playing a key role. While high-penetrance mutations are well-studied, moderate-to-low-penetrance mutations, which present challenges in clinical decision-making and patient outcomes, are less understood. This study explores the quality of life of breast cancer patients with moderate-penetrance mutations, focusing on the psychosocial and physical consequences of mastectomy and reconstruction to improve patient-centered care. **Materials and Methods**. A cohort of 620 breast cancer patients treated at Regina Maria Private Health Network, Bucharest, between January 2022 and July 2024 was identified. From this group, 61 patients were selected based on the following criteria: (1) meeting NCCN genetic testing guidelines, (2) carrying moderate-to-low-penetrance mutations, (3) undergoing bilateral mastectomy with double reconstruction, and (4) agreeing to complete a modified version of the BREAST-Q questionnaire. Genetic testing was performed using a 125-gene next-generation sequencing panel. Statistical analyses included non-parametric tests to examine group differences and correlations. **Results**. Significant correlations were found between several factors. Emotional distress was positively correlated with concerns for family, while couple relationships and financial burden showed a strong positive association. Negative correlations were found between couple relationships and self-concept. Distress levels varied, with “Interference with personal relationships” causing more distress than “Impact on employment”, and financial burden causing more distressing than impact on sexuality. **Conclusions**. Prophylactic mastectomy significantly reduces cancer risk for women with moderate-penetrance mutations. This study highlights the relationship between surgical choices and quality-of-life factors, advancing personalized prevention strategies and emphasizing patient-centered care.

## 1. Introduction

Breast cancer (BC) remains a significant global health concern, particularly for individuals with genetic predispositions. Among these, patients with high-penetrance mutations in genes such as *BRCA1*, *BRCA2*, *CDH1*, *PALB2*, *PTEN*, *STK11*, and *TP53* have been extensively studied, with the literature highlighting a severe impact on their quality of life due to both the disease and the aggressive treatment approaches often required [1]. However, less is known about the quality of life for patients diagnosed with moderate-to-low-penetrance mutations, despite this group facing distinct challenges in risk assessment, clinical decision-making, and post-treatment recovery [1,2].

This article aims to explore the multifaceted impact of surgical treatment on the quality of life among breast cancer patients with moderate-to-low-penetrance genetic mutations. By addressing this knowledge gap, we hope to provide insights into the physical, emotional, and psychosocial outcomes for this subset of patients, ultimately contributing to improved treatment protocols and more personalized, patient-centered care.

## 2. Overview of Hereditary Breast Cancer: Genetic Risk and Preventive Strategies

The most prevalent disease to be diagnosed and the primary cause of cancer-related mortality for women globally is BC. Hereditary genetic mutations in genes such as *BRCA1/2*, *ATM*, *CHEK2*, *PALB2*, and others that are passed down through families account for 5–10% of cases. Instead of being genetically predisposed, environmental and lifestyle factors play a major role in the remaining 90–95% of cases, which are sporadic [1].

The significance of identifying women with germline mutations linked to hereditary breast cancer (HBC) will be discussed in the ensuing subsections. We will also discuss risk assessment for women with moderate-penetrance mutations, discuss the need of screening for these patients, and look at the role of prophylactic mastectomy as a preventive measure for individuals with these genetic variants.

The identification of pathogenic germline mutations in patients is essential for managing cancer predisposition. Important considerations for suggesting genetic testing are highlighted in the National Comprehensive Cancer Network (NCCN) guidelines, such as patient age, and personal and family cancer histories [2,3]. Even though the NCCN promotes testing for genes like *BRCA1/2*, which are linked to BC, there is increasing support for more thorough multigene panel testing to avoid missing nearly half of germline mutations, as some studies have shown when physicians only test for the genes mentioned in the NCCN guidelines [4,5,6,7].

As previously stated, HBC is responsible for up to 10% of cases of BC, and approximately 6% of patients have pathogenic mutations in genes associated with HBC [8,9]. These genes are classified as either high-risk, increasing the risk of cancer more than fourfold, or moderate risk, increasing the risk two to fourfold. It is interesting to note that about one in eight unaffected individuals meet the NCCN testing criteria, suggesting that molecular HBC prevalence differs by ethnicity [10,11].

To address the psychological and medical ramifications for patients and their families, including cancer prevention and treatment options, genetic counseling is essential after an HBC diagnosis [12,13]. Genetic testing for family members should be encouraged by counseling, and risk assessments should be customized for each case. With the help of specialized high-risk clinics that provide coordinated care through multidisciplinary teams, long-term follow-up care for HBC entails a comprehensive approach that includes risk-reducing surgeries, medications, and routine imaging [14].

The quality of life after a mastectomy, whether with or without reconstruction, is crucial, as it significantly impacts overall well-being and emotional health. Many survivors face physical changes, emotional challenges, and concerns about body image, which can influence their quality of life. Addressing these aspects through support systems, therapy, and education can empower individuals to adapt to their new circumstances, fostering resilience and enhancing life satisfaction. Ultimately, ensuring a good quality of life post-mastectomy contributes to holistic healing and helps individuals embrace their identities beyond their diagnosis.

As was mentioned in the previous subsection, HBC risk has been associated with pathogenic variants in genes like *ATM*, *BRCA1*, *BRCA2*, *CHEK2*, *PALB2*, *RAD51C*, and *RAD51D*. Notably, *ATM* and *CHEK2* confer moderate risk, whereas *BRCA1/2* and *PALB2* mutations present a high risk of developing HBC [9,15,16,17]. In the general population, the lifetime risk of BC is about 12%. However, this risk increases significantly for BRCA1/2 carriers. The risks for *BRCA1* and *BRCA2* carriers are 60–66% and 55–61%, respectively, by age 70 [15,18].

For BRCA mutation-carriers, risk-reducing mastectomy (RRM) offers over a 90% reduction in BC risk, and may improve quality of life by lowering the anxiety associated with cancer recurrence. Despite the similar survival rates between RRM and breast-conserving surgery (BCS), studies suggest that contralateral prophylactic mastectomy (CPM) significantly decreases mortality risk for these patients. In contrast, patients with moderate-penetrance mutations make varied surgical choices; some select CPM, while others favor regular imaging surveillance. Factors influencing these decisions include premenopausal status and the specific mutation type. These findings highlight the importance of tailored genetic counseling to support informed, patient-centred choices, and they underscore the need for additional research to assess the long-term outcomes of these decisions [19,20,21,22,23].

## 3. NCCN Guidelines for Genetic Testing and Breast Cancer Screening

Before delving into the main sections of this study, it is important to present the National Comprehensive Cancer Network (NCCN) guidelines, as they formed a crucial part of the inclusion criteria for participants. These guidelines outline recommendations for genetic testing, particularly for those at higher risk of breast cancer due to genetic mutations. In this section, we discuss the general criteria for genetic testing in breast cancer, and highlight the guidelines for screening individuals with high-penetrance and moderate-penetrance mutations. The NCCN guidelines were followed in our study to ensure a standardized and evidence-based approach to participant selection and genetic evaluation. These guidelines not only provide recommendations for testing specific genes, but also suggest appropriate screening protocols to identify individuals at higher risk of developing breast cancer, guiding both early diagnosis and preventive measures.

### 3.1. NCCN Guidelines for Genetic Testing

The NCCN Guidelines, Version 1.2025, were adhered to in our study. The general criteria for genetic testing and the criteria for BC associated with high-penetrance gene mutations will be described in the sections that follow [24].

General genetic testing is recommended in specific clinical scenarios, including the following: when an individual has a blood relative with a known pathogenic or likely pathogenic (P/LP) variant in a cancer-susceptibility gene; when previous limited testing (such as single-gene or testing without deletion/duplication analysis) was negative, and further multigene testing is desired; when a P/LP variant is found in tumor genomic testing with potential germline implications; and when results could inform decisions on systemic therapy or surgical options [24]. The primary barriers to universal testing remain age, personal and family history, and ancestry criteria [25,26]. Studies show that nearly half of patients with pathogenic mutations would not be identified using standard testing criteria, underscoring the need for additional testing options that can yield clinically actionable results [27,28,29]. Economic evaluations suggest that broader testing, especially for women under 50, may be beneficial in reducing the incidence of BC, as well as being cost-effective [8,30].

Genetic testing for genes linked to high-penetrance BC, including *BRCA1*, *BRCA2*, *CDH1*, *PALB2*, *PTEN*, *STK11*, and *TP53*, is recommended for individuals with specific personal or family histories that indicate a higher risk. For those with a personal history of BC, testing is particularly advised if the diagnosis was made before the age of 50. If genetic results could influence treatment choices, such as the potential use of PARP inhibitors or Olaparib, testing is also recommended at any age. Additionally, testing is necessary for TNB, multiple primary BCs, and lobular BC with a family history of diffuse gastric cancer. Men of Ashkenazi Jewish heritage or those with BC are also eligible for genetic testing due to their higher risk profiles [24].

Family history also has a big impact on the need for genetic testing. It is recommended that people get tested if they have close relatives who have been diagnosed with male BC, pancreatic cancer, ovarian cancer, metastatic prostate cancer, or BC before the age of fifty. Further research is necessary because a family history of three or more cases of breast or prostate cancer also suggests a higher genetic risk [24].

Genetic testing is also recommended for those with first- or second-degree relatives who meet these high-risk criteria, or who have a calculated risk of more than 5% for *BRCA* mutations based on predictive models like Tyrer-Cuzick. Those who do not meet these criteria for high-penetrance mutations may still benefit from screening for other hereditary cancer syndromes, or from following general NCCN cancer screening recommendations to effectively manage their risk [24].

The role of genetic testing and counseling in BC treatment is increasingly being recognized, especially in relation to newly identified mutations [31,32]. The American and European continents’ approaches to genetic testing continue to differ significantly [33]. The American Society recommends routine genetic testing for all women with BC, and supports the use of genetic testing to guide surgical decisions [34,35].

### 3.2. NCCN Guidelines for BC Screening in Women with Moderate-Penetrance Genetic Mutations

According to the NCCN Cancer Surveillance Guidelines Version 1.2025 for individuals with moderate-penetrance mutations linked to BC, specific genes warrant increased awareness and proactive screening efforts. These genes, *ATM*, *BARD1*, *CHEK2*, *MSH2*, *MLH1*, *MSH6*, *PMS2*, *EPCAM*, *NF1*, *RAD51C*, and *RAD51D*, are associated with an estimated absolute lifetime BC risk of 20–25%. While this risk is lower than that seen in high-penetrance genes such as *BRCA1/2*, *CDH1*, *PALB2*, or *PTEN*, which have a 50–70% risk range, it still represents a meaningful risk and justifies individualized screening protocols [24].

In Table 1, the NCCN guidelines outline the recommended screening approaches. Annual mammography is recommended, and breast MRI (with or without contrast) is suggested as an additional consideration. The guidelines advise beginning imaging screenings around ages 30–40, depending on the specific gene involved, with adjustments based on family history to determine the optimal start age.

According to American doctors, women between the ages of 40 and 74 should get mammograms every two years, which could result in an additional 20 million screenings [36,37]. However, there are problems with over-screening and the need to find women who are less likely to be at risk. Genetic factors may play a significant role in this identification, as estimates indicate that roughly 1.3 million women may be deemed low risk [18,38].

Current evidence does not support recommending RRM as a routine preventive measure for individuals with moderate-penetrance mutations. Instead, family history should play a key role in guiding RRM discussions between patients and physicians.

It is worth noting that the genes in the group including *MSH2*, *MLH1*, *MSH6*, *PMS2*, and *EPCAM* have no specific recommendations for mammography or breast MRI beyond those that apply generally to other moderate-penetrance genes regarding RRM and family history. Due to the presence of insufficient data for these genes, assessing family history remains the best approach. In cases where there is no family history of hereditary BC linked to these genes, following the general guidelines for other moderate-penetrance genes may be an appropriate course of action.

## 4. Materials and Methods

### 4.1. Selection of the Cohort

We initially identified a cohort of 620 patients who were diagnosed with breast cancer and treated at the Regina Maria Private Health Network in Bucharest, Romania. These patients received their diagnoses and treatments between January 2022 and July 2024, serving as the starting point for our analysis. From this total cohort, a subset was selected based on specific criteria.

Patients were included in the study based on four key criteria, as follows: the first was meeting at least one of the NCCN criteria for genetic testing; the second was presenting a mutation in a gene with moderate-to-low penetrance; the third was undergoing both bilateral mastectomy and double reconstruction; and the fourth was agreeing to participate in the study by completing the quality-of-life questionnaire (see Figure 1).

The 620 BC patients who received oncogenetic consultations met at least one NCCN germline testing criterion were evaluated using a comprehensive next-generation sequencing (NGS) panel encompassing 125 genes. Genetic testing was performed using sequencing data from the Illumina platform (Illumina Inc., San Diego, CA, USA), aligned to the human reference genome (GRCh37/hg19). Variants were analyzed using gnomAD and ClinVar databases, along with specialized software tools. Variant calling was performed using GATK (version 4.3.0.0, Broad Institute, Cambridge, MA, USA), while annotation and interpretation were conducted with VarSeq (version 2.4.0, Golden Helix, Bozeman, MT, USA) and Alamut Visual (version 1.11, SOPHiA GENETICS, Rolle, Switzerland). Copy number variations (CNVs) were detected using ExomeDepth (version 1.1.15, University of Cambridge, Cambridge, UK). Pathogenicity was evaluated by a clinical team following ACMG guidelines, and significant findings were confirmed using Sanger sequencing with the ProDye® Terminator Sequencing System (Promega Corporation, Madison, WI, USA) and through Sanger sequencing services provided by Eurofins Genomics (Ebersberg, Germany).

To statistically analyze the quality of life in a specific subset of patients, we selected individuals from a total cohort of 620 breast cancer patients treated at the Regina Maria Private Health Network in Bucharest, Romania. The selection criteria focused on patients diagnosed with moderate-penetrance gene mutations who had undergone mastectomy followed by reconstruction. Ultimately, only 61 patients met the criteria, having moderate-to-low-penetrance genetic mutations and undergoing contralateral prophylactic mastectomy with bilateral reconstruction.

The figure below illustrates the patient selection process for this study. It outlines the step-by-step criteria applied to narrow down the initial cohort of 620 breast cancer patients to the final subset of 61 patients. Each stage of selection is detailed, from meeting NCCN criteria for genetic testing to identifying those with moderate-to-low-penetrance mutations, undergoing bilateral mastectomy with reconstruction, and agreeing to participate by completing the quality-of-life questionnaire. This visual representation highlights the systematic approach used to ensure a focused and relevant study group (see Figure 1).

### 4.2. Explanation of Questionnaire and Distress Measurement

In our study, we used a shortened version of the BREAST-Q quality of life questionnaire (Version 2.0, 2017, developed by a team of researchers led by Dr. Anne F. Klassen, a professor at McMaster University in Hamilton, ON, Canada), designed to assess key aspects of both the mastectomy and reconstruction processes. The full BREAST-Q covers a wide range of domains relevant to the patient experience, including adverse effects from radiation, cancer worry, fatigue, impact on work, and various physical and psychosocial aspects. For mastectomy, the full set of items includes categories such as chest-related physical concerns, psychosocial impact, return to activity, and sexual well-being. For reconstruction, it adds specific concerns about abdominal, back, and shoulder physical effects, animation deformity, and more detailed psychosocial issues related to the changes in appearance and sexuality.

While the full BREAST-Q consists of 56 items, our study aimed to focus on the most critical factors affecting quality of life for these patients while minimizing participant burden. We selected questions that cover the core areas of concern identified in the original questionnaire, such as physical changes (e.g., appearance and functionality), psychosocial impact (e.g., self-concept, personal relationships), and distress caused by the surgery and diagnosis. By grouping these essential items, we sought to capture the most pertinent aspects of the patients’ experiences related to their genetic diagnosis, mastectomy, and reconstruction, while also ensuring that the questionnaire remained manageable for completion in a clinical setting.

In addition to covering the core domains, we specifically aimed to measure the levels of distress patients felt throughout the process. Our questions addressed a range of distressors, including emotional responses to the genetic diagnosis, fear related to surgery, changes in physical appearance, impacts on personal and couple relationships, concerns about family members, financial burdens, and the effect on sexuality. These questions were carefully chosen to assess the intensity of distress patients experienced at various stages—from diagnosis through recovery—further complementing the more general quality of life measures in the full BREAST-Q. Each item was rated on a 1-to-5 scale, where 1 represented very low distress, 2 low distress, 3 moderate distress, 4 high distress, and 5 very high distress. The specific questions used are listed below (see Table 2).

These questions were designed to cover the key emotional and practical impacts that are central to the patient experience during this process, ensuring that we comprehensively capture the levels of distress related to each stage of diagnosis, surgery, and recovery. By using these specific items, we aimed to balance both quality of life and distress measurement, providing a more nuanced understanding of the challenges faced by this patient population.

### 4.3. Applied Statistical Methods

Descriptive analyses were conducted on these factors to understand the general trends. Furthermore, we investigated potential positive or negative correlations between these factors by applying statistical methods to calculate the significance of pairwise associations. To determine whether the queried factors differed significantly across subgroups defined by types of surgery, reconstruction, and complications, we tested for normality in the distribution of responses within each subgroup. Non-parametric one-way ANOVA tests were then applied to identify statistically significant differences in the questionnaire results when split by surgical type, reconstruction type, or complication type for each assessed quality-of-life factor. We also conducted an analysis to determine whether certain factors assessed in the questionnaire were perceived as more significant than others. To achieve this, we applied the Wilcoxon test to all possible pairwise combinations of factors in the questionnaire to identify any statistically significant differences between the medians of the data for each assessed factor. Statistical analyses were conducted using Jamovi (version 2.6.17, The Jamovi Project, Sydney, Australia). Figures were created using Microsoft® Excel for Mac (version 16.83, Microsoft Corporation, Redmond, WA, USA).

## 5. Description of the Study Group

In this section, we will present a detailed overview of the characteristics of the final study group. This includes information on the patients’ age, the genetic mutations they carry, as well as the histological types and subtypes of breast cancer identified. Additionally, we will describe the mastectomy procedures performed, including the techniques used for both the initial surgery and the prophylactic mastectomy for the contralateral breast. Furthermore, we will outline the various reconstructive procedures utilized in the study group, providing a comprehensive understanding of the treatments and patient profiles involved.

Regarding the demographics and mutation frequencies, the average age at diagnosis for the final study group consisting of 61 patients was 48.6 years. The distribution of mutations among the cohort was as follows: *CHEK2*—21 patients (29.2%), *ATM*—11 patients (15.2%), *PMS2*—8 patients (11.1%), *MLH1*—7 patients (9.7%). Other mutations included *RAD50* (2.7%), *NF1* (1.4%), *NBN* (4.1%), and various others, accounting for the remaining cases (see Figure 2).

Histological characteristics—The most prevalent histological type identified was ductal invasive carcinoma (50 patients, 82%), followed by lobular invasive carcinoma (8 patients, 14%), and other rare histological subtypes like medullary and papillary (3 patients, 3%) (see Figure 3). The majority were classified as Luminal B (44 patients, 73%), followed by Luminal A (7 patients, 11%) and triple-negative BC (TNB) (10 patients, 16%) (see Figure 4).

All 61 patients included in the study underwent double mastectomy, with a prophylactic mastectomy performed on the contralateral breast. The two techniques used for mastectomy were skin-sparing mastectomy (SSM) (20 patients, 33%) and nipple-sparing mastectomy (NSM) (41 patients, 67%) (see Figure 5). For the reconstructive procedures, three techniques were employed, as follows: DIEP (Deep Inferior Epigastric Perforator flap) (15 patients, 25%), PPD (prepectoral reconstruction with dermal matrix) (20 patients, 32%), and SP (subpectoral reconstruction) (26 patients, 43%) (see Figure 6).

## 6. Quality-of-Life Questionnaire Results and Statistical Methods Applied

We performed a statistical analysis of the data, calculating the mean, standard error of the mean, standard deviation, 95% confidence interval, and variance. Additionally, we conducted the Shapiro–Wilk test to assess whether the data provided by the patients followed a normal distribution, which would guide the selection of appropriate tests and coefficients for normal or non-normal distributions. The results of the test show that all subgroups for each aspect assessed did not follow a normal distribution, with *p*-values below 0.05. Although the W coefficient was close to 1, suggesting normality in some data subgroups, the overall data did not follow a normal distribution.

### 6.1. Descriptive Analysis

The data collected and the responses to the questionnaire are presented in more detail in Table 3, which includes all the values for each subgroup of questions.

### 6.2. Correlation Analysis

We aimed to determine whether the levels of distress for each question assessed in the quality-of-life questionnaire are positively or negatively correlated with each other. To achieve this, we calculated the Spearman correlation coefficient and the *p*-value for all pairs of factors. However, not all combinations of factors presented *p*-values < 0.05. The combinations that demonstrated statistically significant correlations (*p*-value < 0.05) are presented in Table 4, where we also include the value of Spearman’s rho for each significant pair.

### 6.3. Results of Quality-of-Life Factors by Subtype of Nominal Variables

Since the data for the factors assessed in the questionnaire, grouped according to the subtypes of nominal variables (type of surgery and type of reconstruction), did not follow a normal distribution (*p* < 0.05), we applied the One-Way ANOVA non-parametric (Kruskal–Wallis) test. As a result, there was no need to calculate the *p*-values for the homogeneity of variances within these subgroups using Levene’s test.

Subsequently, we conducted a one-way ANOVA (non-parametric) (Kruskal–Wallis) test to analye the factors assessed in the questionnaire (Fear related to genetic diagnosis, Age, Fear related to surgery, Change in appearance, Change in self-concept, Interference with personal relationships, Impact on sexuality, Impact on employment, Impact on couple relationships, Concern related to daughters or relatives, Overall emotional distress, and Financial burden) in relation to variables such as the type of surgical intervention and type of reconstruction.

Using the types of surgery, types of reconstruction, and types of complications within the group, the One-Way ANOVA (non-parametric) (Kruskal–Wallis) test showed that there were no statistically significant differences between any two subgroups based on the types of surgery, types of reconstruction, or types of complications, regardless of the factors assessed in the questionnaire. The overall *p*-values can be found in Table 5.

### 6.4. Perceived Significance of Quality-of-Life Factors

We wanted also to determine whether certain factors in the quality-of-life questionnaire were perceived as more significant than others by the participants, in the sense of identifying whether the subjects experienced greater distress in relation to one aspect compared to another. To achieve this, we conducted the Wilcoxon signed-rank test, as the data for the parameters measured were not normally distributed, which precluded the use of the paired Student’s *t*-test.

After comparing all the factors assessed in the questionnaire, as well as all combinations of two factors, the analysis revealed several statistically significant differences in the level of distress experienced with respect to each element (Table 6). However, it is important to note that the other comparisons between pairs of factors did not yield *p*-values less than 0.05. Thus, the level of distress experienced in these other associations did not present statistically significant differences. In other words, the distress levels associated with the remaining factor pairings were found to be comparable, and there was no strong evidence to suggest that these factors influenced distress in a distinct manner.

## 7. Discussion

In this discussion section, we will delve into the key findings from the study, starting with an analysis of the identified correlations in the quality-of-life questionnaire results, which provide valuable insights into the factors influencing the well-being of breast cancer patients. We will also explore the perceived significance of these quality-of-life factors, assessing how patients prioritize various aspects of their health and well-being. The role of tumor boards in breast cancer management will be examined, highlighting their importance in delivering comprehensive, patient-centered care that considers both the physical and emotional aspects of treatment. Finally, we will review the overall quality of life for patients who have undergone prophylactic mastectomy, synthesizing the findings to offer a holistic understanding of the impact of this preventive surgical approach. Each of these subsections will contribute to a broader discussion of how treatment decisions, healthcare teams, and patient experiences intersect to shape the journey of breast cancer patients.

### 7.1. Analysis of Identified Correlations in Quality-of-Life Questionnaire Results

The statistical analysis of the quality-of-life questionnaire reveal several significant positive and negative correlations between the assessed factors, shedding light on the interplay of various aspects of life among the studied patients. These correlations provide valuable insights into the ways emotional, relational, and practical challenges influence each other. Importantly, only the correlations discussed below demonstrated statistical significance (*p* < 0.05). All other possible pairwise correlations between the factors did not reach statistical significance, making the ones presented here the only meaningful correlations identified in this study.

A positive correlation was observed between concern related to daughters or relatives and overall emotional distress (Spearman’s rho = 0.266, *p* = 0.038). This finding suggests that as concerns for the well-being of daughters or other relatives increase, patients experience greater overall emotional distress. This relationship is understandable, as the health and future of family members, particularly daughters, often constitute a significant source of anxiety for individuals managing health challenges. Similarly, there was a positive correlation between the impact on couple relationships and fear related to surgery (Spearman’s rho = 0.270, *p* = 0.035), indicating that greater fear of surgical procedures tends to exacerbate the strain on couple dynamics. The anxiety and stress associated with surgery likely intensify dependence and emotional challenges, placing additional pressure on relationships.

A strong positive correlation was also found between the impact on couple relationships and financial burden (Spearman’s rho = 0.755, *p* < 0.001). This indicates that financial stress is intricately tied to difficulties within couple relationships. Healthcare costs and other financial burdens are well-documented sources of relational conflict, amplifying the emotional strain experienced by couples in such situations. Another positive correlation emerged between the impact on employment and interference with personal relationships (Spearman’s rho =0.513, *p* < 0.001). This suggests that difficulties in maintaining employment due to health-related challenges are closely linked to disruptions in personal relationships. Employment-related stress may spill over into personal dynamics, affecting interpersonal connections and further compounding the challenges faced by patients.

Conversely, negative correlations reveal instances where certain factors inversely influence one another. A negative correlation was found between the impact on couple relationships and change in self-concept (Spearman’s rho = −0.261, *p* = 0.042), suggesting that individuals who report greater relational strain tend to experience fewer positive changes in their self-concept. This could be due to the relational challenges undermining their capacity for self-reflection and growth. Another negative correlation was observed between the impact on couple relationships and concern related to daughters or relatives (Spearman’s rho= −0.282, *p* = 0.028). This finding suggests that those with higher family concerns report fewer difficulties within their couple relationships. It is possible that focusing on family concerns fosters a sense of unity and shared purpose, mitigating relational strain.

Additionally, a negative correlation was identified between change in self-concept and fear related to surgery (Spearman’s rho= −0.367, *p* = 0.004). This indicates that patients who undergo a positive change in self-concept report lower levels of fear related to surgery. This relationship may suggest that individuals with a stronger sense of personal growth or adaptability are better equipped to handle the psychological challenges of undergoing surgical procedures.

Finally, a negative correlation was identified between the impact on sexuality and age (Spearman’s rho= −0.345, *p* = 0.006), indicating that as age increases, the perceived impact on sexuality decreases. This may reflect a shift in priorities or expectations regarding sexual activity among older patients, for whom this aspect of life might hold less central importance compared to younger individuals.

In summary, the analysis showed that only these correlations exhibited statistical significance (*p* < 0.05). All other potential pairwise correlations between the factors did not demonstrate significance, emphasizing that the relationships presented here are the only meaningful findings in this study. These results highlight the intricate interplay between psychological, relational, and demographic factors in this patient population. Positive correlations reveal how certain factors, such as financial burdens and relational difficulties, tend to intensify one another, while negative correlations suggest compensatory mechanisms, such as improved self-concept reducing surgical fear. These findings provide valuable insights for targeted interventions, helping address specific vulnerabilities like emotional distress, relational challenges, and financial strain.

### 7.2. Analysis of Perceived Significance of Quality-of-Life Factors

In this section, we will discuss the results previously presented, focusing on the comparisons between various factors assessed in the quality-of-life questionnaire. Specifically, we will explore the significant differences observed in distress levels related to “Interference with personal relationships”, “Impact on employment”, “Financial burden”, and “Impact on sexuality”. By analyzing the median and mean scores, we aim to better understand how participants perceive and experience distress in these areas. These discussions provide insights into which factors were considered more significant or impactful, highlighting the nuanced emotional challenges faced by the participants.

In the comparison between “Interference with personal relationships” and “Impact on employment”, a significant difference was found (*p* = 0.04). Both factors had the same median value of 3.00, but their mean values differed slightly. The mean for interference with personal relationships was 3.43, while the mean for the impact on employment was 3.07. This suggests that, on average, participants reported slightly more distress related to personal relationships than to employment issues. Although the medians were the same, the mean scores indicate a subtle but consistent difference in how distress manifests in these two areas.

A second significant difference was observed between “Financial burden” and “Impact on sexuality” (*p* = 0.022). Here, the financial burden had a significantly higher median of 4.00 compared to the median of 3.00 for the impact on sexuality. The mean scores also supported this finding, with a mean of 3.52 for financial burden and 3.13 for the impact on sexuality. These results suggest that, on average, participants experienced more distress related to financial difficulties than to issues surrounding sexuality. The clear difference in both median and mean scores points to a stronger emotional impact from financial stress.

Lastly, the comparison between “Financial burden” and “Impact on employment” showed a statistically significant difference (*p* = 0.049). The financial burden again had a higher median of 4.00, compared to 3.00 for the impact on employment. The mean values were 3.52 for financial burden and 3.07 for employment impact, reinforcing the finding that financial difficulties were perceived to cause more distress than employment-related issues. The results imply that financial concerns are a more significant source of distress than employment issues, as reflected in both the median and mean scores.

### 7.3. The Importance of Tumor Boards in BC Management

The most prevalent cancer in women, BC, claimed 685,000 lives globally in 2020 and caused 2.3 million new cases. By the end of that year, 7.8 million women had received a BC diagnosis in the preceding five years. Important risk factors for BC include age, reproductive and hormonal factors, metabolic and dietary issues, previous thoracic radiation therapy, BC or dysplasia, family history, and genetic predisposition. Increased screening and awareness have led to more early-stage diagnoses, which have improved five-year survival rates and resulted in more effective conservative surgical treatments [39,40,41].

Effective breast cancer treatment requires a complex, multidisciplinary approach, as patient care involves a variety of healthcare professionals, such as surgeons and psychotherapists. Examining the role of tumor boards in relation to studies on the quality of life of breast cancer patients is essential for enhancing patient care and outcomes. Structured healthcare is essential for improved patient outcomes and care coordination. As a result, the US, Asia, Australia, Europe, and the UK now frequently use a multidisciplinary approach. This approach addresses the complexity of clinical judgments and is crucial in oncology [39,42].

Multidisciplinary tumor boards (MTBs), which are composed of teams of experts who collaborate to determine the best course of treatment for patients, are the gold standard for cancer treatment. This model enhances clinical outcomes, optimizes the use of available resources, and enhances the training of healthcare professionals by promoting shared accountability in diagnosis, treatment planning, and evaluation [43,44].

In many countries, multidisciplinary care is formally recognized as essential to the management of BC, and is used as a standard for accreditation and funding. Multidisciplinary BC care has been the subject of extensive research, but implementation has recently drawn increased interest. However, due to differing definitions and contexts, the exact effectiveness of multidisciplinary care is still unknown [39,42,43,44].

The studies show that the use of MTB has improved diagnosis, especially in the areas of imaging and pathology interpretation. Following MTB discussions, patient recommendations regarding treatment often changed, leading to a decrease in prophylactic mastectomies and an increase in immediate breast reconstructions. While the studies’ survival rates differed, some reported improvements following the implementation of MTB, while others showed no discernible changes [39,45,46].

Patients who participated in MTBs had a statistically significantly—14%—lower risk of dying than those who did not, according to a pooled analysis of three high-quality studies. Small sample sizes, a lack of randomized clinical trials, and a restricted number of eligible studies were some of the review’s drawbacks. Half of the included studies were rated as having “fair” methodological quality, and the majority were observational, which makes them susceptible to bias. Additionally, only studies published in English or Italian were included in the review, which may have left out pertinent data in other languages [39,47,48,49].

### 7.4. Overview of Quality of Life After Undergoing a Prophylactic Mastectomy

Prophylactic mastectomy symbolizes a crucial decision in a woman’s existence, which can have serious physical, emotional, social, and psychological outcomes, responsible in many cases for a deterioration in the quality of life [50,51,52].

The psychological impact of mastectomy is multifaceted, involving complex emotional responses and significant adjustments in body image, self-perception, and interpersonal relationships. The extent and nature of the procedure may vary widely among patients, but research consistently emphasizes a range of psychological repercussions [53]. As of now, there are still not enough studies regarding this major change that could be perceived as an inciting psychological incident by the great majority of patients. Even though it was believed that a prophylactic mastectomy would be the ideal solution for reducing the level of anxiety in patients with an elevated risk of primary BC or a BC recurrence, it was observed that fear of cancer recurrence was a concern in 84.2% of the cases, shortly after the surgical intervention [54]. To diminish the level of stress and anxiety, a great number of patients considered to have a relatively low risk of developing a breast neoplasm (absence of *BRCA* mutation or relevant family history) opt for a contralateral prophylactic mastectomy, even though many studies showed no benefits over other less invasive surgeries. This phenomenon is known as “surgeons operating on anxiety”. All this notwithstanding, instead of relying on statistical data, the patients consider their risk to be higher if they have a pessimistic affect toward BC, resulting in CPM as a choice for their apparent well-being [55]. In some cases, regular follow-ups could alleviate this concern to a certain extent, yet many women report persistent health and cancer anxiety [56].

In relation to physical impacts, mastectomy may contribute to a profound sense of loss. Patients often suffer from disturbing emotions, including sadness, anger, grief, and regret, all of them leading to sleep disturbances as well [57,58]. Aside from the scar, which is an obvious characteristic, a percentage greater than 50% reported retraction, stiffness, tightness, adherence, and a reduction in the range of arm motion. On the contrary, some women tend to associate the scar with a victory reminding them that they are alive, and they survived the cancer. Most women reported coming to terms with the inevitability of scarring, while others did not share this sentiment. Factors such as expectations regarding scarring, satisfaction with care, the quality of information provided, age, the significance placed on the scar, changes over time, and coping strategies appear to affect the degree of acceptance [58].

Moreover, the sexual life of individuals following mastectomy can undergo a substantial transformation. The most frequently incriminated factors included a deterioration in body image, along with a reduction in breast sensitivity and bodily sensations, diminished feelings of sexual attractiveness, and decreased sexual desire [59]. Many studies report that sexual contentment receives the worst mean score out of the entire BREAST-Q questionnaire [52,60,61,62], with up to 9 in 10 women presenting with sexual dysfunction.

Nonetheless, being diagnosed with breast cancer at an early age poses significant obstacles, in terms of both therapy and future reproductive prospects. Ovarian function can be severely compromised by hormone therapy, radiation, and chemotherapy, underscoring the urgent need to preserve fertility. Patients can protect their reproductive autonomy by cryopreserving oocytes, embryos, or ovarian tissue before therapy. It is crucial to have timely conversations about fertility preservation, but access to these services is frequently hampered by a lack of understanding and cost limitations. To support informed decision-making, oncofertility counseling must be incorporated into cancer care. Additionally, fertility preservation has been linked to better quality of life, less psychological discomfort, and a higher chance of pregnancy after treatment, thanks to developments in assisted reproductive technology [63,64].

However, recent studies challenge the practice of offering ovarian tissue cryopreservation (OTC) to breast cancer patients, suggesting that its efficacy and utility are overestimated. They argue that while oocyte vitrification may benefit psychological well-being, fertility potential after chemotherapy, particularly in women under 40, is often disregarded. Besides this, low success rates of OTC have led to many unused frozen ovarian tissues. The authors call for a re-evaluation of OTC as a fertility preservation option in favor of more effective methods [65,66].

Taking into consideration the diversity of feelings a woman can encounter after such a procedure, psychotherapy would be the best long-term treatment option. The patients may learn to process their trauma, address their loss and grieving, and learn new ways to cope with anxiety and depression. In addition to that, follow-up meetings with the oncologist should be routinely effectuated.

However, a crucial aspect to consider in healthcare is patient-centered care (PCC), which emphasizes the inclusion of patients’ individual preferences, needs, and values in clinical decision-making. In breast cancer treatment, particularly concerning surgical options, PCC plays a crucial role in enhancing patient satisfaction and outcomes. In the study of Ostapenko E et al., 2022, a machine learning algorithm was used to predict individual patient-reported outcomes at a one-year follow-up, and the process was validated. This tool aims to facilitate personalized, patient-centered decision-making for women undergoing mastectomy and reconstruction, highlighting the importance of tailoring treatment plans to individual expectations and circumstances [67].

Despite its recognized importance, the implementation of PCC in breast cancer treatment varies significantly. A scoping analysis revealed diverse interpretations of PCC, suggesting that embracing this heterogeneity and applying PCC as an umbrella term for all healthcare that acknowledges the person in the patient may be beneficial. This perspective encourages healthcare providers to focus on the individual experiences and values of patients when planning and delivering care [68].

Furthermore, patient-centered communication has been shown to positively impact cancer patients’ perceptions of care quality, self-efficacy, and trust in their doctors. Engaging patients in their care through effective communication fosters better understanding, satisfaction, and adherence to treatment plans, underscoring the significance of PCC in clinical practice [69].

Incorporating PCC into breast cancer surgery involves not only considering the clinical aspects of treatment, but also addressing the psychological, emotional, and social dimensions of patient care. By prioritizing individualized care and shared decision-making, healthcare providers can improve patient outcomes and satisfaction in breast cancer treatment. Taking into consideration the diversity of feelings a woman can encounter after such a procedure, psychotherapy would be the best long-term treatment option. The patients may learn to process their trauma, address their loss and grieving, and learn new ways to cope with anxiety and depression. In addition to that, follow-up meetings with the oncologist should be routinely effectuated.

## 8. Future Implications and Directions for Research

This study examines the impacts of moderate-to-low-penetrance gene mutations on the quality of life for patients undergoing reconstruction and preventative mastectomy. It enhances our understanding of the psychological and physical effects of these procedures, and highlights important future considerations for patient care, clinical practice, and research.

The findings provide a foundation for advancements centered on patients, emphasizing personalized decision-making, mental health support, long-term outcome monitoring, and equitable access to healthcare. As surgical and genetic technologies continue to evolve, ongoing research will be essential for refining preventive oncology strategies and improving patient well-being.

In the following subsections, we will explore some specific future implications of this study in greater depth.

### 8.1. Personalized Surgical Decision-Making

A more individualised approach to surgical decision-making is one of the study’s most important ramifications. The differences found in quality-of-life characteristics imply that more informed surgical decisions may result from customised risk–benefit calculations that consider a patient’s genetic predisposition, psychological fortitude, and personal beliefs. To help patients navigate their treatment options, future research should investigate the creation of decision-support tools that integrate psychological well-being measurements, patient-reported results, and genetic risk assessment.

### 8.2. Enhancing Psychosocial Support and Mental Health Interventions

The necessity for integrated psychological support services is highlighted by the substantial association that exists between emotional suffering and elements like interpersonal pressure, financial load, and body image issues. To improve patient well-being, future healthcare models should give priority to mental health therapies such as peer support groups, post-mastectomy psychotherapy, and preoperative counseling. The creation of focused mental health solutions may benefit from longitudinal research evaluating the long-term psychological effects of mastectomy and reconstruction.

### 8.3. Expanding Research on Moderate-Penetrance Mutations

Although high-penetrance genes like BRCA1/2 have been the subject of much research, there is a rising desire to learn more about moderate-penetrance mutations. Gene–environment interactions, the cumulative effects of several moderate-to-low-risk variants, and their influence on clinical decision-making should all be examined in future genetic research. Comprehensive genetic research that incorporates multi-omics data may yield more accurate risk evaluations and help guide customized screening and preventative measures.

### 8.4. Long-Term Outcomes of Prophylactic Mastectomy and Reconstruction

Although the immediate quality-of-life impacts of mastectomy and reconstruction are highlighted in this study, the long-term repercussions are still not well studied. Prospective cohort studies that follow patients over long periods of time would yield important information on things like chronic pain, surgery satisfaction, and changing psychosocial adaptations. By comprehending these long-term patterns, surgical methods and postoperative care guidelines may be improved.

### 8.5. Integration of Multidisciplinary Tumour Boards in Patient Care

Considering the complex relationships between genetic risk factors, surgical options, and psychosocial outcomes, incorporating multidisciplinary tumour boards (MTBs) into standard clinical practice could improve patient-centered care. Future research should evaluate how MTBs influence surgical decision-making, treatment adherence, and overall patient satisfaction. This could lead to the development of standardized protocols aimed at optimizing treatment outcomes.

## 9. Limitations of This Study

This section outlines factors that may influence the study’s findings and generalizability. While we have here provided valuable insights into the quality of life of breast cancer patients undergoing prophylactic mastectomy and reconstruction, certain limitations should be considered:Small sample size—The final cohort of 61 patients, reduced from an initial 620, may limit generalizability;Selection bias—The study focuses only on patients with moderate-to-low-penetrance gene mutations, excluding broader breast cancer populations;Self-reported data—Reliance on patient questionnaires introduces potential biases related to memory recall and subjective interpretation;Shortened questionnaire—A modified BREAST-Q was used for better response rates, possibly omitting relevant quality-of-life domains;Limited scope—Emotional distress and physical appearance were emphasized, leaving out factors like long-term health outcomes and social support;No control group—The absence of a comparison group limits conclusions on the impact of prophylactic mastectomy;Geographical constraints—Conducted at a single clinic in Bucharest, the findings of this study may not apply to other regions or healthcare settings;Genetic profile exclusion—Patients with other genetic mutations were not included, possibly overlooking different quality-of-life experiences;Confounding variables—Factors like socioeconomic status, psychological support, and surgery timing were not fully accounted for;Limited follow-up—The study lacks long-term assessment, restricting insights into the lasting effects of surgery on quality of life.

By acknowledging these limitations, we provide a critical perspective on the findings and highlight areas for future research to enhance the understanding of patient experiences.

## 10. Conclusions

To summarize, this study provides a thorough outline of the complex effects of treating breast cancer, with an emphasis on the quality of life for patients undergoing a preventative mastectomy. Complex relationships between emotional anguish, relational difficulties, financial pressures, and the psychological and physical effects of surgery were found through the examination of quality-of-life questionnaire data. These results highlight how crucial it is to address both the physical and relational aspects of patient care, in addition to the emotional ones. The necessity of cooperative, patient-centered methods that consider both clinical and psychosocial aspects in decision-making is further highlighted by the function of multidisciplinary tumor boards in the treatment of breast cancer. Furthermore, the analysis of quality of life following a prophylactic mastectomy reveals the significant physical and psychological changes that patients go through, underscoring the need for continued assistance, such as psychotherapy and routine follow-up care, to lessen the long-term effects of surgery. In the end, this study offers insightful information that can guide future research, clinical procedures, and patient care plans targeted at enhancing the quality of life for patients with breast cancer.

## Figures and Tables

**Figure 1 jcm-14-01140-f001:**
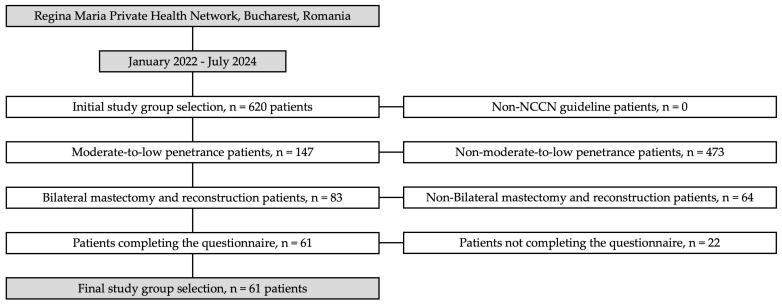
Patient selection flowchart. The figure below outlines the patient selection process for this study. From an initial cohort of 620 breast cancer patients, exclusions were made based on four criteria, as follows: not meeting NCCN genetic testing guidelines, not having moderate-to-low-penetrance mutations, not undergoing bilateral mastectomy with reconstruction, and not consenting to complete the quality-of-life questionnaire. The final study group included 61 patients. The flowchart visually represents the selection process and the number of patients excluded at each stage.

**Figure 2 jcm-14-01140-f002:**
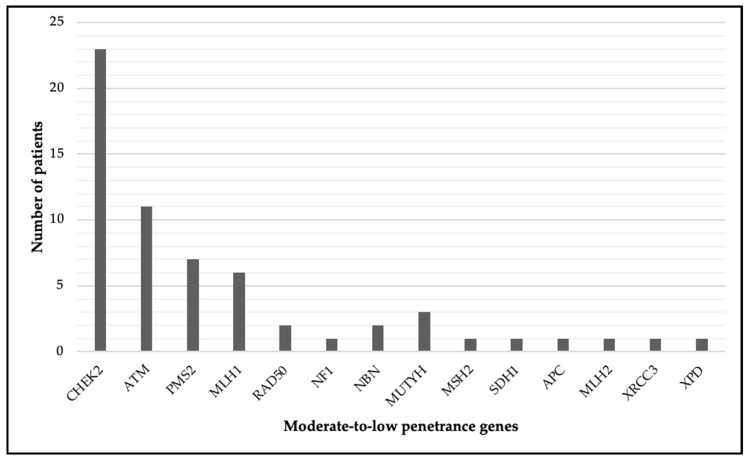
Distribution of patients by gene mutation. The figure below shows a clustered column chart depicting the distribution of patients based on the specific genes in which they presented mutations. Each column represents the number of patients with mutations in different genes, providing a clear overview of the genetic variation within the study group.

**Figure 3 jcm-14-01140-f003:**
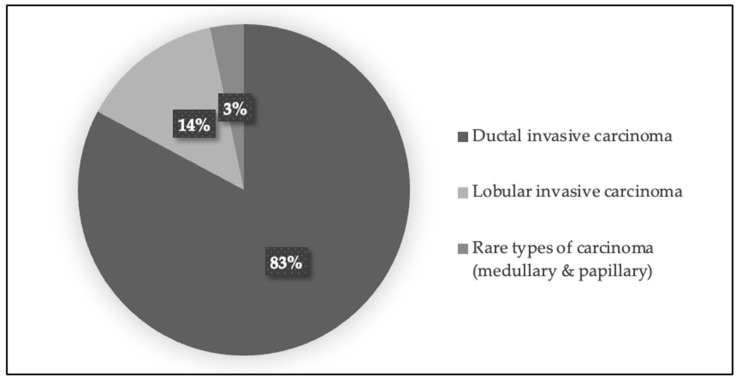
Distribution of patients by histological tumor type. The figure below presents a pie chart illustrating the percentage distribution of patients based on the histological types of their tumors. Each segment represents a different tumor type, providing a clear visual breakdown of the histological characteristics within the study group.

**Figure 4 jcm-14-01140-f004:**
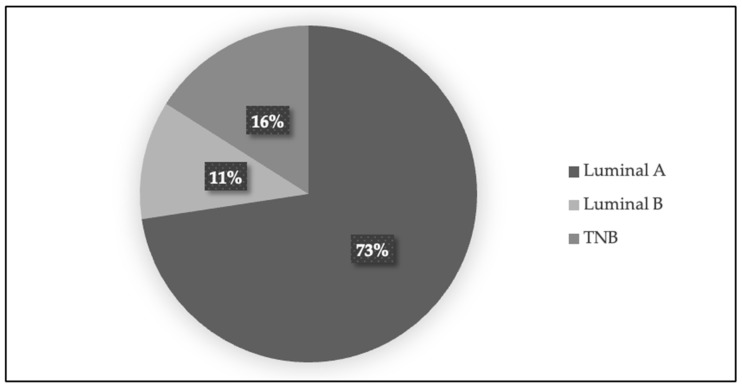
Distribution of patients by breast cancer subtype. The figure below presents a pie chart illustrating the percentage distribution of patients based on the breast cancer subtypes. Each segment represents a specific subtype, such as Luminal A, Luminal B, and Triple Negative, providing a visual breakdown of the breast cancer characteristics within the study group.

**Figure 5 jcm-14-01140-f005:**
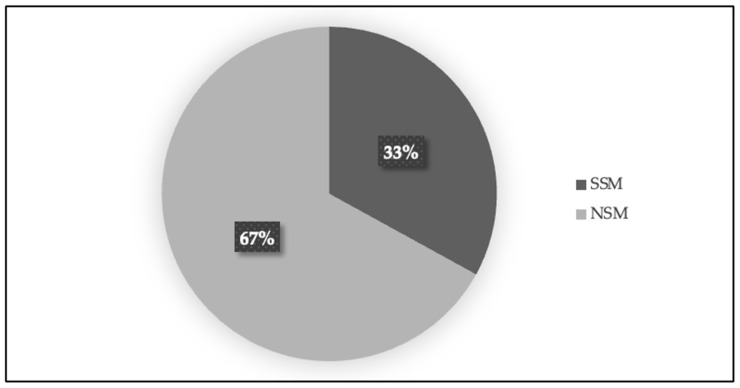
Distribution of patients by mastectomy technique. The figure below presents a pie chart illustrating the percentage distribution of patients based on the two mastectomy techniques used: skin-sparing mastectomy (SSM) and nipple-sparing mastectomy (NSM). Each segment represents the proportion of patients who underwent each technique in the study group.

**Figure 6 jcm-14-01140-f006:**
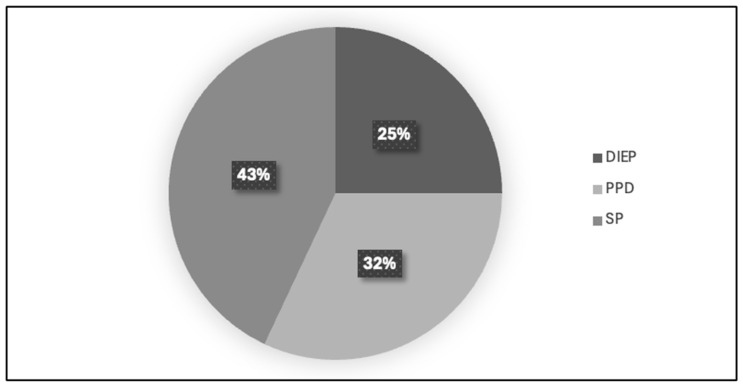
Distribution of patients by reconstructive procedure technique. The figure below presents a pie chart illustrating the percentage distribution of patients based on the three reconstructive techniques used, as follows: DIEP (Deep Inferior Epigastric Perforator flap), PPD (prepectoral reconstruction with dermal matrix), and SP (subpectoral reconstruction). Each segment represents the proportion of patients who underwent each technique in the study group.

**Table 1 jcm-14-01140-t001:** NCCN Cancer Surveillance Guidelines Version 1.2025 for individuals with moderate-to-low-penetrance mutations for BC.

Gene	Screening			Risk Reduction	Observations
	Imagistic Method	Peridiocity	From Age (Years)		
*ATM*	Mammogram ^1^	Annual	40	Evidence insufficient for risk-reducing mastectomy	Management based on family history
	Breast MRI ± contrast ^2^	n/a ^3^	30–35		
*BARD1*	Mammogram	Annual	40		
	Breast MRI ± contrast	n/a			
*CHEK2*	Mammogram	Annual	40		
	Breast MRI ± contrast	n/a	30–35		
*MSH2*, *MLH1*, *MSH6*, *PMS2*, *EPCAM*	Mammogram	n/a	n/a		
	Breast MRI ± contrast				
*NF1*	Mammogram	Annual	30		
	Breast MRI ± contrast	n/a	30–35		
*RAD51C*	Mammogram	Annual	40		
	Breast MRI ± contrast	n/a			
*RAD51D*	Mammogram	Annual	40		
	Breast MRI ± contrast	n/a			

^1^. A mammogram is recommended according to NCCN guidelines. ^2^. Breast MRI, with or without contrast, is suggested for consideration in line with NCCN guidelines. ^3^. There are no available data in the NCCN guidelines.

**Table 2 jcm-14-01140-t002:** Table of questions used to assess quality of life in the study.

No. Crt.	Question
1.	How much distress has the genetic diagnosis caused you? (Fear related to genetic diagnosis)
2.	How much distress have the bilateral mastectomies and reconstruction surgeries caused you? (Fear related to surgery)
3.	How much distress did the time between receiving the diagnosis and achieving full recovery after the reconstruction surgery cause in your physical appearance? (Change in appearance)
4.	How much distress did the time between receiving the diagnosis and achieving full recovery after the reconstruction surgery impact your self-concept? (Change in self-concept)
5.	How much has the surgery or diagnosis interfered with your personal relationships, and how distressing has this been? (Interference with personal relationship)
6.	How much distress did the time between receiving the diagnosis and achieving full recovery after the reconstruction surgery cause in your couple relationship? (Impact on couple relationship)
7.	How much distress did the time between receiving the diagnosis and achieving full recovery after the reconstruction surgery have on your employment? (Impact on employment)
8.	How much distress did the time between receiving the diagnosis and achieving full recovery after the reconstruction surgery cause in your concerns for your daughters or other relatives? (Concern related to daughters or relatives)
9.	How much distress did the time between receiving the diagnosis and achieving full recovery after the reconstruction surgery cause in terms of financial burden? (Financial burden)
10.	How much distress did the time between receiving the diagnosis and achieving full recovery after the reconstruction surgery cause in relation to your sexuality? (Impact on sexuality)
11.	How much overall emotional distress have you experienced related to the diagnosis and treatment? (Overall emotional distress)

**Table 3 jcm-14-01140-t003:** Descriptive analysis of the statistical data obtained from the questionnaire.

	Mean	SE	95% Confidence Interval		Median	SD	Variance	Shapiro–Wilk Test	
			Lower	Upper				W	*p*
Impact on couple relationship	3.43	0.143	3.14	3.71	4.00	1.117	1.249	0.898	<0.001
Age	49.72	1.011	47.70	51.74	49.00	7.893	62.304	0.950	0.014
Financial burden	3.52	0.138	3.25	3.80	4.00	1.074	1.154	0.889	<0.001
Concern related to daughters or relatives	3.18	0.154	2.87	3.49	3.00	1.204	1.450	0.904	<0.001
Overall emotional distress	3.25	0.121	3.00	3.49	3.00	0.943	0.889	0.848	<0.001
Impact on employment	3.07	0.173	2.72	3.41	3.00	1.352	1.829	0.880	<0.001
Fear related to surgery	3.34	0.132	3.08	3.61	3.00	1.031	1.063	0.908	<0.001
Impact on sexuality	3.13	0.137	2.86	3.41	3.00	1.072	1.149	0.909	<0.001
Change in appearance	3.48	0.136	3.20	3.75	4.00	1.058	1.120	0.896	<0.001
Fear related to genetic diagnosis	3.31	0.137	3.04	3.59	3.00	1.073	1.151	0.901	<0.001
Interference with personal relationship	3.43	0.158	3.11	3.74	3.00	1.231	1.515	0.878	<0.001
Change in self-concept	3.30	0.143	3.01	3.58	3.00	1.116	1.245	0.904	<0.001

**Table 4 jcm-14-01140-t004:** Statistically significant correlations between distress levels for quality-of-life questionnaire factors.

Factor 1	Factor 2	Spearman’s Rho	*p*-Value
Change in self-concept	Fear related to surgery	−0.367	0.004
Concern related to daughters or relatives	Overall emotional distress	0.266	0.038
Impact on couple relationship	Fear related to surgery	0.270	0.035
	Financial burden	0.755	<0.001
	Change in self-concept	−0.261	0.042
	Concern related to daughters or relatives	−0.282	0.028
Interference with personal relationship	Fear related to surgery	0.261	0.042
Impact on employment	Interference with personal relationship	0.513	<0.001
Impact on sexuality	Age	−0.345	0.006

**Table 5 jcm-14-01140-t005:** One way ANOVA (non-parametric) (Kruskal–Wallis) test.

	Type of Surgery as Grouping Variable		Type of Recontruction as Grouping Variable	
	χ^2^	*p*	χ^2^	*p*
Age	1.5078	0.471	0.797	0.671
Impact on sexuality	3.1430	0.208	0.486	0.784
Financial burden	2.1572	0.340	1.082	0.582
Concern related to daughters or relatives	2.6104	0.271	0.855	0.652
Overall emotional distress	5.5218	0.063	0.652	0.722
Impact on couple relationship	1.3007	0.522	0.432	0.806
Impact on employment	2.5222	0.283	0.447	0.800
Fear related to genetic diagnosis	0.6163	0.735	2.033	0.362
Fear related to surgery	1.7785	0.411	1.293	0.524
Interference with personal relationship	0.9873	0.610	3.136	0.208
Change in appearance	0.0990	0.952	2.208	0.332
Change in self-concept	3.1153	0.211	0.365	0.833

**Table 6 jcm-14-01140-t006:** Wilcoxon test *p*-values for distress levels across paired factor comparisons in the quality-of-life questionnaire.

First Factor	Second Factor	Wilcoxon Test *p*-Value
Interference with personal relationship	Impact on employment	0.040
Impact on sexuality	Financial burden	0.022
Impact on employment		0.049

## Data Availability

Data are contained within the article (see Appendix A).

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
