# Peer review of "Quality-of-Life Assessment in Patients Undergoing Mastectomy and Breast Reconstruction for Moderate-Penetrance Gene-Related Breast Cancer"

_jcm, 2025, doi:10.3390/jcm14041140_

Round 1

Reviewer 1 Report

Comments and Suggestions for Authors

The manuscript presented for review is an interesting analysis of the impact of prophilactic mastectomy and subsequent reconstruction in breast cancer patients with low-to-moderate penetrance genetic mutations.

The idea is original an tries to bring light to an existing knowledge gap. I read the manuscript with interest and I found it informative and well written.

My only concern about this manuscript is that I believe that the authors should provide a more extensive explanation on why they chose to use a reduced version of the questionaire and why they believe this reduction does not influence the validity of that questionaire. 

Author Response

  1. Summary

Thank you very much for taking the time to review this manuscript. Please find the detailed responses below and the corresponding revisions and corrections highlighted in the re-submitted files.

  1. Point-by-point response to Comments and Suggestions for Authors

Comment 1. The manuscript presented for review is an interesting analysis of the impact of prophilactic mastectomy and subsequent reconstruction in breast cancer patients with low-to-moderate penetrance genetic mutations.

Response 1. Thank you for pointing this out.

Comment 2. The idea is original an tries to bring light to an existing knowledge gap. I read the manuscript with interest and I found it informative and well written.

Response 2. Thank you for your kind and encouraging feedback. We greatly appreciate your recognition of the originality of the idea and the effort to address an existing knowledge gap. We are pleased that you found the manuscript informative and well-written.

Comment 3. My only concern about this manuscript is that I believe that the authors should provide a more extensive explanation on why they chose to use a reduced version of the questionaire and why they believe this reduction does not influence the validity of that questionaire.

Response 3. Thank you for your valuable feedback. We acknowledge the importance of justifying our choice to use a shortened version of the BREAST-Q questionnaire and addressing its potential impact on validity.

The decision to reduce the questionnaire was made to enhance response rates and minimize participant burden while still capturing the most critical aspects of quality of life relevant to our study population. Given the length of the full BREAST-Q, we selected the most important item groups from the original questionnaire to ensure a comprehensive yet concise assessment. Our goal was to maintain the integrity of the questionnaire while improving completion rates, particularly since it was administered in a clinical setting, where patient time and engagement are key factors.

By reducing the number of items, we aimed to ensure that patients responded to all questions, preventing incomplete responses and enhancing the reliability of the collected data. This approach allowed us to gather meaningful insights while optimizing the time required for completion.

To address this concern more explicitly, we revised the Methods section to provide a more detailed justification of our approach, including the specific criteria used for selecting the most important item groups and acknowledging the potential limitations of using a reduced questionnaire. Please see the section “4.2. Explanation of Questionnaire and Distress Measurement” (lines 250 – 289)

Reviewer 2 Report

Comments and Suggestions for Authors

First of all, I would like to thank the Editor for the opportunity to review this manuscript. This is an original article aimed at analyzing the quality of life of patients with mutations in medium-to-low penetrance genes for breast cancer. The Authors considered 61 breast cancer patients who underwent surgical treatment at a single center. The article addresses a highly relevant and timely topic.

However, some clarifications are necessary:

  1. The questionnaire used is not the BREAST-Q but a simplified version. Is this the first time it has been used? Are there any previous studies in the literature that have utilized this questionnaire?
  2. The patients underwent bilateral mastectomy with reconstruction using either DIEP or implants. Did all patients undergoing prepectoral reconstruction use a matrix? If so, could the Authors explain why?
  3. The Authors rightly emphasize the importance of the MDM. Could you clarify whether these patients were considered for cryopreservation or referred for a specific gynecological evaluation?
  4. Could you outline the potential future implications of your study?
  5. Listing the study’s limitations is appropriate, but this section could be more concise.

Additionally, I would like to suggest the following references:

Ostapenko E, Nixdorf L, Devyatko Y, Exner R, Wimmer K, Fitzal F. Prepectoral Versus Subpectoral Implant-Based Breast Reconstruction: A Systemic Review and Meta-analysis. Ann Surg Oncol. 2023 Jan;30(1):126-136. doi: 10.1245/s10434-022-12567-0. Epub 2022 Oct 16. PMID: 36245049; PMCID: PMC9726796.

Franceschini G, Scardina L, Di Leone A, Terribile DA, Sanchez AM, Magno S, D'Archi S, Franco A, Mason EJ, Carnassale B, Murando F, Orlandi A, Barone Adesi L, Visconti G, Salgarello M, Masetti R. Immediate Prosthetic Breast Reconstruction after Nipple-Sparing Mastectomy: Traditional Subpectoral Technique versus Direct-to-Implant Prepectoral Reconstruction without Acellular Dermal Matrix. J Pers Med. 2021 Feb 22;11(2):153. doi: 10.3390/jpm11020153. PMID: 33671712; PMCID: PMC7926428.

Magnoni F, Sacchini V, Veronesi P, Bianchi B, Bottazzoli E, Tagliaferri V, Mazzotta E, Castelnovo G, Deguidi G, Rossi EMC, Corso G. Surgical Management of Inherited Breast Cancer: Role of Breast-Conserving Surgery. Cancers (Basel). 2022 Jul 1;14(13):3245. doi: 10.3390/cancers14133245. PMID: 35805017; PMCID: PMC9265273.

Author Response

  1. Summary

Thank you very much for taking the time to review this manuscript. Please find the detailed responses below and the corresponding revisions and corrections highlighted in the re-submitted files.

  1. Point-by-point response to Comments and Suggestions for Authors

Comment 1. First of all, I would like to thank the Editor for the opportunity to review this manuscript. This is an original article aimed at analyzing the quality of life of patients with mutations in medium-to-low penetrance genes for breast cancer. The Authors considered 61 breast cancer patients who underwent surgical treatment at a single center. The article addresses a highly relevant and timely topic.

Response 1. Thank you very much for your thoughtful review and for recognizing the originality and relevance of our article. We appreciate the opportunity to address such an important topic and are glad to hear that you find it timely and relevant. Your feedback is valuable, and we are grateful for your careful consideration of the manuscript.

Comment 2. The questionnaire used is not the BREAST-Q but a simplified version. Is this the first time it has been used? Are there any previous studies in the literature that have utilized this questionnaire?

Response 2. Thank you for your insightful question. The questionnaire used in our study was specifically created for this research, and to our knowledge, it has not been previously utilized in other studies. However, the items we assessed align with those explored in other studies using different validated instruments. For example, Deshpande et al. (2022) (doi: 10.7759/cureus.27703) employed the QOL-BC questionnaire, which evaluates similar domains such as physical and psychological well-being, distress, fear, and social well-being.

Additionally, our study specifically aimed to analyze distress levels, which differentiates our approach. To provide further clarity, we have refined Section 4: Materials and Methods by introducing a new subsection, 4.2. Explanation of Questionnaire and Distress Measurement, where we elaborate on the development of our questionnaire and its focus on distress evaluation. (lines 250 – 289)

We appreciate your feedback and hope these revisions address your concerns.

Comment 3. The patients underwent bilateral mastectomy with reconstruction using either DIEP or implants. Did all patients undergoing prepectoral reconstruction use a matrix? If so, could the Authors explain why?

Response 3. Thank you for your thoughtful question. Among the patients who underwent prepectoral reconstruction, some received a matrix (either acellular dermal matrix or synthetic mesh), while others did not. The decision to use or omit a matrix was based on individual case factors, including the need for additional support, cosmetic outcomes, and the goal of reducing complications.

We did not provide extensive details on surgical techniques in the manuscript because our primary focus was on quality of life outcomes based on the type of surgery performed. Additionally, given our small sample size (DIEP: 15 patients, 25%; PPD: 20 patients, 32%; SP: 26 patients, 43%), further stratification would have significantly reduced the statistical power of our analysis.

However, we conducted a One-way ANOVA (non-parametric Kruskal-Wallis) test, which demonstrated no statistically significant differences in quality-of-life outcomes based on surgery type, or reconstruction method. Since these factors did not significantly impact the questionnaire results, we did not expand on surgical details in the manuscript.

We appreciate your inquiry and hope this explanation clarifies our approach.

Comment 4. The Authors rightly emphasize the importance of the MDM. Could you clarify whether these patients were considered for cryopreservation or referred for a specific gynecological evaluation?

Response 4. Thank you for your insightful question. In our study, the youngest patient at the time of diagnosis was 38 years old, and most patients were either perimenopausal or post/menopausal at the time of diagnosis and treatment. Among those who were still of reproductive age, none expressed a desire to have children. As a result, none of the patients were referred to a gynecologist for cryopreservation discussions.

However, we acknowledge that our original manuscript did not address this topic. To enhance the discussion on quality of life in breast cancer patients, we have added two brief paragraphs addressing cryopreservation considerations. Nevertheless, we have not expanded further on this topic, as fertility preservation was not a focus of this study, and assessing distress related to reproduction was not relevant given that none of the patients desired future pregnancies. (lines 629 – 646)

We appreciate your feedback and hope this clarification provides the necessary context.

Comment 5. Could you outline the potential future implications of your study?

Response 5. Thank you for your suggestion. In response to your request, we have incorporated a dedicated section, "Future Implications and Directions for Research" (Section 8), which outlines the broader potential future implications of our study. This section delves deeper into several key areas, including:

  • Personalized Surgical Decision-Making
  • Enhancing Psychosocial Support and Mental Health Interventions
  • Expanding Research on Moderate-Penetrance Mutations
  • Long-Term Outcomes of Prophylactic Mastectomy and Reconstruction
  • Integration of Multidisciplinary Tumour Boards in Patient Care

We hope this expanded section provides a comprehensive overview of the future directions for research in this area. Thank you again for your helpful input. (lines 681 – 736)

Comment 6. Listing the study’s limitations is appropriate, but this section could be more concise.

Response 6. Thank you for your helpful feedback. In response to your suggestion, we have made the Limitations section more concise. We hope this revision meets your expectations and improves the manuscript’s overall readability. Thank you again for your constructive comment. (lines 738 – 763)

Comment 7. Additionally, I would like to suggest the following references:

- Ostapenko E, Nixdorf L, Devyatko Y, Exner R, Wimmer K, Fitzal F. Prepectoral Versus Subpectoral Implant-Based Breast Reconstruction: A Systemic Review and Meta-analysis. Ann Surg Oncol. 2023 Jan;30(1):126-136. doi: 10.1245/s10434-022-12567-0. Epub 2022 Oct 16. PMID: 36245049; PMCID: PMC9726796.

- Franceschini G, Scardina L, Di Leone A, Terribile DA, Sanchez AM, Magno S, D'Archi S, Franco A, Mason EJ, Carnassale B, Murando F, Orlandi A, Barone Adesi L, Visconti G, Salgarello M, Masetti R. Immediate Prosthetic Breast Reconstruction after Nipple-Sparing Mastectomy: Traditional Subpectoral Technique versus Direct-to-Implant Prepectoral Reconstruction without Acellular Dermal Matrix. J Pers Med. 2021 Feb 22;11(2):153. doi: 10.3390/jpm11020153. PMID: 33671712; PMCID: PMC7926428.

- Magnoni F, Sacchini V, Veronesi P, Bianchi B, Bottazzoli E, Tagliaferri V, Mazzotta E, Castelnovo G, Deguidi G, Rossi EMC, Corso G. Surgical Management of Inherited Breast Cancer: Role of Breast-Conserving Surgery. Cancers (Basel). 2022 Jul 1;14(13):3245. doi: 10.3390/cancers14133245. PMID: 35805017; PMCID: PMC9265273.

Response 7. Thank you for suggesting the additional references. In response, we have incorporated the following key points into Section 7.4: Overview of Quality of Life after Undergoing a Prophylactic Mastectomy:

  • Psychotherapy and Follow-Up Care: We emphasized the importance of psychotherapy as a long-term treatment option to help patients process trauma, loss, and cope with anxiety and depression, alongside regular follow-up meetings with oncologists.
  • Patient-Centered Care (PCC): We discussed the significance of PCC in breast cancer treatment, highlighting its role in personalized decision-making and the positive impact of effective patient-provider communication on patient satisfaction and outcomes.

We hope these additions adequately reflect the references you provided and further enrich the manuscript. Thank you again for your valuable input. (lines 647 – 680)